# Ordinal Cochran-Mantel-Haenszel Testing and Nonparametric Analysis of Variance: Competing Methodologies

J. C. W. Rayner [1,2,*] and G. C. Livingston, Jr. [2]

1    National Institute for Applied Statistics Research Australia, University of Wollongong, Wollongong, NSW 2522, Australia
2    Centre for Computer-Assisted Research Mathematics and its Applications, School of Information and Physical Sciences, University of Newcastle, Newcastle, NSW 2308, Australia
*    Correspondence: john.rayner@newcastle.edu.au

**Abstract:** The Cochran-Mantel-Haenszel (CMH) and nonparametric analysis of variance (NP ANOVA) methodologies are both sets of tests for categorical response data. The latter are competitor tests for the ordinal CMH tests in which the response variable is necessarily ordinal; the treatment variable may be either ordinal or nominal. The CMH mean score test seeks to detect mean treatment differences, while the CMH correlation test assesses ordinary or (1, 1) generalized correlation. Since the corresponding nonparametric ANOVA tests assess arbitrary univariate and bivariate moments, the ordinal CMH tests have been extended to enable a fuller comparison. The CMH tests are conditional tests, assuming that certain marginal totals in the data table are known. They have been extended to have unconditional analogues. The NP ANOVA tests are unconditional. Here, we give a brief overview of both methodologies to address the question "which methodology is preferable?".

**Keywords:** correlation test; Friedman test; generalized correlations; Kruskal-Wallis test; mean score test

## 1. Introduction

In the following we describe the essential features of the Cochran-Mantel-Haenszel (CMH) and the more recent nonparametric analysis of variance (NP ANOVA) methodologies. Both analyze categorical response data, so which is to be preferred? For greater detail the reader is referred to Rayner and Livingston Jr [1]. We recommend implementing the tests using the R package CMHNPA in Livingston Jr and Rayner [2]. See https://cran.r-project.org/web/packages/CMHNPA/index.html, accessed on 3 October 2022.

For that reason in the following we do not give explicit formulae.

Section 2 describes the basic CMH tests and Section 3 describes NP ANOVA. The extended CMH tests and unconditional CMH analogues are met in both Sections 2 and 4. A brief conclusion is given in Section 5. A data set relating job satisfaction and income in males and females demonstrates features as they are encountered.

## 2. The Basic CMH Tests

The basic CMH methodology consists of four tests of association when given categorical count data for a very specific design. The counts are $\{N_{ihj}\}$ in which $N_{ihj}$ is the number of times the $i$th of $t$ treatments falls in the $h$th of $c$ categories on the $j$th of $b$ blocks or strata. We call this the *CMH design*. The completely randomized and randomized block designs are CMH designs, while the balanced incomplete block and Latin squares designs are not.

The CMH tests are designated as tests for

- overall partial association (OPA)
- general association (GA)

- mean scores (MS) and
- correlation (C).

The CMH OPA test assesses *overall* association while the CMH GA test seeks to detect an *average* association between the treatment and response variables. The CMH MS test seeks to detect treatment mean differences between the responses and the CMH C test seeks to detect linear association between the treatment and response variables.

Under the null hypothesis of no association the CMH test statistics have asymptotic $\chi^2$ distributions, with degrees of freedom that reflect the focus of the tests. As we progress from the OPA to the GA to the MS and finally to the C test, the degrees of freedom are $b(c-1)(t-1)$, $(c-1)(t-1)$, $(t-1)$ and 1 respectively. The tests with larger degrees of freedom are more *omnibus*, detecting general alternatives with moderate power. As the degrees of freedom decrease, the tests become more *focused*, having greater power in the reduced parameter space but being insensitive to alternatives outside of that space. See the related discussion in Rayner et al. ([3] page 3). Thus, for detecting correlation effects the CMH C test will have greater power than CMH GA test, but it will have less power than the CMH GA test for other alternatives. See, for example, the discussion in Rayner et al. ([3] sections 5.3 and 5.4). In this reference the focus is on the power of the Pearson goodness of fit test based on $X_P^2$ and the tests based on its components; several examples are given.

When the responses are ranks, the CMH MS for the completely randomized design is the Kruskal-Wallis test, while that for the randomized block design is the Friedman test.

The genesis of the CMH methodology are papers by Cochran [4], who proposed a test for average partial association, and by Mantel and Haenszel [5], whose main interest was in *retrospective studies*, rather than *prospective studies*. In a CMH design the category totals within strata, $\sum_i N_{ihj}$ are all known constants. An appropriate model for constructing CMH test statistics is thus the extended hypergeometric distribution model. Inference is *conditional* on the marginals taking their observed values. Conditional inference is a valid inference paradigm, but one that no doubt is unfamiliar to many data analysts. In prospective studies the category totals within strata are not known before the analysis; they are random variables.

Rayner and Best [6] constructed unconditional analogues of the CMH tests to explore possible differences in inference when both conditional and unconditional CMH tests were applied to the same data sets. Of most concern in conditional analyses is that the counts in each response category are assumed to be known before analyzing the data. In practice the differences between corresponding conditional and unconditional tests are usually small: *p*-values tend to be similar.

To implement the CMH MS test *scores* must be assigned to the response categories, while to implement the CMH C test scores must be assigned to both the response categories and to the treatments. *Natural* scores (1, 2, 3, . . . ) are frequently chosen, but note the data driven choices made in the job satisfaction example following. Category midpoints or medians also commonly used, as these also reflect the data.

The OPA and GA CMH tests that require no scores are referred to as *nominal* tests. The MS and C CMH tests that do require scores are referred to as *ordinal* tests. For the nominal CMH tests there are Pearson tests that are competitor tests. They test the same hypotheses as the nominal CMH tests and their test statistics have the same asymptotic chi-squared sampling distributions. The competitor Pearson test for the CMH GA test first aggregates the data over strata. Software to analyze data using the nominal CMH tests may not be readily available to many users, and we hope that our CMHNPA package will fill the gap. For the competitor Pearson test for the CMH GA the R function chisq.test is available. Writing R code for the competitor Pearson test for the CMH OPA should not be difficult for R coders with a little experience.

*Job satisfaction example*. The data in Table 1, from Agresti [7], relate job satisfaction and income in males and females. It derives from the 1996 General Social Survey conducted by the US National Opinion Research Center and hence is not artificial data. Gender induces two strata; the treatments are income with categories scored 3, 10, 20 and 35, while the

response is job satisfaction with categories scored 1, 3, 4 and 5. Agresti reports *p*-values for the CMH GA, MS and C tests to be 0.3345, 0.0288 and 0.0131 respectively. We find the CMH OPA test has *p*-value 0.3005. At the 0.05 level there is neither overall partial nor general association. At the same level there is evidence the job satisfaction means differ, and evidence of a correlation: as income increases so does job satisfaction.

**Table 1.** Satisfaction and income in males and females.

| Gender | Income ($1000) | Job Satisfaction | | | | |
| | | Very Dissatisfied | A Little Satisfied | Moderately Satisfied | Very Satisfied | Total |
|---|---|---|---|---|---|---|
| Female | <5 | 1 | 3 | 11 | 2 | 17 |
| | 5 to 15 | 2 | 3 | 17 | 3 | 25 |
| | 15 to 25 | 0 | 1 | 8 | 5 | 14 |
| | >25 | 0 | 2 | 4 | 2 | 8 |
| | Total Female | 3 | 9 | 40 | 12 | 64 |
| Male | <5 | 1 | 1 | 2 | 1 | 5 |
| | 5 to 15 | 0 | 3 | 5 | 1 | 9 |
| | 15 to 25 | 0 | 0 | 7 | 3 | 10 |
| | >25 | 0 | 1 | 9 | 6 | 16 |
| | Total Male | 1 | 5 | 23 | 11 | 40 |

Using the natural scores 1, 2, 3, 4 for both treatment and response scores does not affect the nominal tests that do not use scores. Instead of the CMH MS *p*-value of 0.0288 we find 0.0264; instead of the CMH C *p*-value of 0.0131 we find 0.0101. In our experience provided the scores used are 'sensible', the *p*-values are similar. Sensible scores include data driven choices such as class midpoints and medians, and natural scores.

For the nominal tests the corresponding unconditional tests are Pearson tests with *p*-values 0.2777 corresponding to the OPA test and 0.2415 corresponding to the GA test: numerically close to the conditional *p*-values. Both approaches reach the same conclusions.

For the ordinal tests corresponding to the CMH MS test *p*-value of 0.0288 the unconditional test has *p*-value 0.0129. Corresponding to the CMH C test *p*-value of 0.0131 the unconditional test has *p*-value 0.0103. Again the *p*-values are numerically close.

## 3. NP ANOVA

In the context of multifactor ANOVA the nonparametric analysis of variance (NP ANOVA) methodology was developed in Rayner and Best [8,9]. It was then extended to designs consistent with the general linear model in Rayner [10]. The approach used in developing these tests is to construct a table of counts relating the categorical responses to the independent variables. Partitions of the Pearson statistic are then used to identify component random variables. Particular components may be used as data in the ANOVA of interest. Smooth product multinomial models are constructed and tests for the smooth parameters of these models given. The smooth parameters are simply related to the ANOVA parameters showing that the tests for the smooth parameters are equivalent to tests for the ANOVA parameters. This brief outline is intended to demonstrate two things about the NP ANOVA approach:

- first, although ANOVA responses are typically continuous, the NP ANOVA methodology assumes categorical responses;
- second, the tests assume only weak smooth models. Assumptions such as normality of the ANOVA residuals are not necessary: inference could be based on resampling tests.

NP ANOVA treats factors for which the levels are not ordered differently from factors for which the levels are ordered. However, in both approaches orthonormal polynomials

are required. The weight functions used are the proportions of distinct outcomes for responses and the proportions of treatment levels. The former will be $\{1/n\}$ if there all $n$ outcomes are different. That will not be the case if, for example, the responses are ranks and ties occur. The construction of such orthonormal functions is discussed in, for example, Rayner et al. [11].

In the implementation of the *unordered* NP ANOVA orthonormal polynomials are constructed on the responses and the intended ANOVA is applied to the data transformed by the first few orthonormal polynomials. These ANOVAs are, in a sense, uncorrelated, and so the conclusions from one ANOVA do not affect the conclusions from any other. The analysis of the data transformed by the orthonormal polynomial of degree $r$ assesses a combination of the moments up to the $r$th for consistency across treatments. If the first $r - 1$ analyses of a particular factor do not reject the null hypothesis of no treatment effects, then the $r$th analysis assesses consistency across treatments of the $r$th moments of that factor. In practice we usually work with $r = 3$ unless there is reason to not do so. See the examples in Rayner and Best [12].

It is of interest to note that, in a sense, the Kruskal–Wallis and Friedman tests are each equivalent to the unordered NP ANOVA first degree tests.

When factors are ordered, we need the concept of *generalized correlations*. Suppose we have bivariate discrete random variables $(X, Y)$. The functions in $\{a_u(X)\}$ are orthonormal with respect to the distribution of $X$ and the functions in $\{b_v(Y)\}$ are orthonormal with respect to the distribution of $Y$. Then, $E[a_u(X) b_v(Y)]$ is the population bivariate generalized correlation of degree $(u, v)$. It is possible to define generalized correlations for arbitrary multivariate distributions. See, for example, Rayner and Beh [13]. One of the insightful properties of generalized correlations is that a necessary and sufficient condition for independence is that all generalized correlations are zero.

For *ordered* NP ANOVA we *first* test whether or not each of the $(u, v)$th generalized correlations is zero, usually for $u$ and $v = 1, 2$ and 3. For bivariate count $\{N_{ij}\}$ with category scores $\{x_i\}$ and $\{y_i\}$ and $\{a_u(X)\}$ and $\{b_v(Y)\}$ orthonormal polynomials put

$$V_{uv} = \sum_i \sum_j N_{ij} a_u(x_i) b_v(y_j) / \sqrt{n}.$$

These $V_{uv}$ are asymptotically standard normal. For a particular couple $(u, v)$ if the $V_{uv}$s are consistent with normality we typically use a t-test to assess whether the generalized correlation is zero. Otherwise the Wilcoxon signed rank test is used. Unless there is good reason to do otherwise, if they all exist we usually test nine generalized correlations, $u$ and $v = 1, 2$ and 3. The $V_{uv} / \sqrt{n}$ are correlations in the usual sense: a covariance divided by the square root of the product of the marginal variances. The $(1, 1)$th generalized correlation is the well-known Pearson correlation while the $(1, 2)$th and $(2, 1)$th generalized correlations are umbrella statistics, assessing such effects as, as the treatment levels increase do the responses increase and then decrease or, conversely, decrease then increase? More complex generalized correlations are usually difficult to interpret.

*Second*, in ordered NP ANOVA we also test whether the chosen generalized correlations vary across levels of sets of the independent variables. To do so orthonormal polynomials are constructed on the responses and on the factors that have ordered levels. This results in multiple sets of orthonormal polynomials. A new response variable for a modified ANOVA is then constructed. The new response is the product of the $u$th orthonormal polynomial on the response, the $v$th orthonormal polynomial on the first ordered factor, . . . , and the $w$th orthonormal polynomial on the last ordered factor. The modified ANOVA is the intended ANOVA modified by omitting the factors that have ordered levels, since these have been accounted for in the new response.

As, for example, in the Job Satisfaction example following, we are proposing performing several tests on the same data set, and this would usually be regarded as exploratory data analysis. That being the case, if the assumptions underpinning the ANOVA are dubious and calculating resampling $p$-values is inconvenient, then the analyst can nevertheless use the ANOVA $p$-values calculated using F distributions and take comfort in the knowl-

edge that ANOVA is broadly robust. See, for example, Ito [14]. Where we have calculated both, the ANOVA F test and resampling *p*-values are generally in good agreement. It is also worth mentioning that even if all null hypotheses are true, in performing several tests at the 0.05 level of significance, roughly 5% of them will reject the null hypothesis. Both points should be born in mind in interpreting the testing.

*Job satisfaction example continued*. To apply the unordered NP ANOVA, we first noted that a two factor ANOVA with interaction resulted in a non-significant interaction term. Subsequently a model without interaction was applied resulting in no significant gender effects and a significant first order income effect. There were no second and third degree income effects. The income *p*-values for the first to third degree analyses are 0.0261, 0.9000 and 0.9074. The Shapiro–Wilk normality *p*-values are all less than 0.01, but we rely on the robustness of the ANOVA.

Looking a little deeper, the gender means are 13.4531 for females and 21.6250 for males. The means for income levels are 6.50000 for less than $5000, 14.07143 for $5000 to $15,000, 16.09524 for $15,000 to $25,000, and 21.26087 for above $25,000. Without any analysis there appears to be a linear trend.

Ordered nonparametric ANOVA was used to test the null hypothesis that the generalized correlations up to the (3, 3)th are zero. The Wilcoxon signed rank test was used because in each case the data were found to have Shapiro–Wilk normality test *p*-values of less than 0.0001. In testing for the degree (1, 1), (1, 2), (1, 3), (2, 1), (2, 2), (2, 3), (3, 1), (3, 2) and (3, 3) generalized correlations the *p*-values were 0.0341, 0.5178, 0.8327, 0.1881, 0.9017, 0.1028, 0.3443, 0.8837 and 0.4018 respectively. The only generalized correlation to give evidence of not being zero is that of order (1, 1). There is some evidence that job satisfaction and income increase together. The ordered NP ANOVA design, a one factor ANOVA with factor gender, has a *p*-value of 0.1935, so there is no evidence that the (1, 1) generalized correlation varies between males and females.

It seems the CMH GA test is not statistically significant because it is testing for too general an alternative: namely generalized correlations up to degree (3, 3). The statistically insignificant higher degree generalized correlations mask the statistically significant (1, 1) generalized correlation. Similarly the CMH GA test could be viewed as assessing univariate moment effects up to degree three, with the second and third order effects masking the first order effect.

## 4. The Extended CMH Tests

In seeking to compare the ordinal CMH and NP ANOVA tests an immediate issue is that the CMH MS test only tests for mean differences, whereas the unordered NP ANOVA can also be used to test for higher univariate moment differences. Similarly the CMH C test assesses only whether the (1, 1)th correlation is zero, whereas NP ANOVA also tests for higher degree generalized correlations. To address this deficiency in the CMH tests, Rayner and Best [12] extended the definitions of the ordinal CMH tests so that they too, could assess univariate moment differences beyond the first, and bivariate generalized correlations beyond the (1, 1)th.

We have noted previously that Rayner and Best [6] developed unconditional analogues of the ordinal CMH tests. However, those tests included unconditional analogues of the CMH MS test extended to detect higher univariate moment differences, and unconditional analogues of the CMH C test extended to detect higher bivariate moment differences. Now the NP ANOVA tests are also unconditional tests. However, these test for quite different univariate moments effects, and test for bivariate moment effects quite differently than the extended CMH correlation tests and their unconditional analogues.

It is of interest to directly compare the conditional and unconditional tests for univariate and bivariate moment differences, and to compare these with the NP ANOVA tests. In the examples we have analyzed differences in the inference between the corresponding tests are usually small. However, this need not be the case. See the example in Best and Rayner (2017) [15] in which a data set is analyzed using Cochran's *Q* test. The unconditional

test is significant at the 0.05 level, whereas the conditional test is not. The data analyst needs to be aware of the paradigms involved and communicate the conclusions carefully.

*Job satisfaction example continued*. The CMH MS test had *p*-value 0.0288, while the extension to test for second moment effect has *p*-value 0.8981. The corresponding unconditional analogues had *p*-values of 0.0129 and 0.6381. We previously noted the NP ANOVA income *p*-values for the first to third degree analyses are 0.0261, 0.9000 and 0.9074. All analyses conclude there are mean differences but no higher moment effects.

The CMH C test had *p*-value 0.0131 while the extension to test for (1, 2), (2, 1) and (2, 2) effects have *p*-values 0.7903, 0.2176 and 0.8897. The *p*-values for the corresponding unconditional analogues are 0.0103, 0.9826, 0.1324 and 0.6288. The corresponding NP ANOVA *p*-values 0.034, 0.518, 0.188 and 0.902 respectively. These are tests for generalized correlations based on the Wilcoxon test. All analyses conclude there are ordinary or (1, 1) generalized correlation effects but no higher order bivariate moment effects.

Overall there is a very satisfactory agreement between the conditional, unconditional and nonparametric ANOVA analyses.

## 5. Conclusions

For the data sets we have analyzed by both extended CMH and NP ANOVA, the differences are small. To undertake competitive size and power studies would be a considerable undertaking because there are many tests to compare and many coherent scenarios. However, from the data analysis we have experience with, we suspect neither set of tests would be superior to the other and any differences will be small.

The major differences between the methodologies are immediate from their construction.

- The CMH designs are but a subset of the fixed effect ANOVAs to which the NP ANOVA apply. For example, the latter apply to the balanced incomplete block and Latin square designs, while ordinal CMH cannot.
- Both the extended CMH correlation tests and NP ANOVA assess generalized correlation effects across the entire data set. However, NP ANOVA also assesses whether these effects vary across blocks or strata.
- The paradigm of inference for the CMH methodology is conditional inference. Many users of statistics are not familiar with conditional inference, and inference is conditional on the assumptions made. While the different paradigms do not often result in different statistical conclusions, that can occur.

Recent work has extended the CMH methodology by adding a suite of unconditional tests and by adding to the MS test tests for higher univariate effects, and adding to the C test tests for higher degree generalized correlation effects. However, the major deficiency is the limited designs to which the CMH tests apply. Nevertheless, for some users it may be important to use the CMH tests, in which case the recent extensions are a useful addition to the suite of tests.

**Author Contributions:** Conceptualization, J.C.W.R.; methodology, J.C.W.R. and G.C.L.J.; software, G.C.L.J.; validation, J.C.W.R. and G.C.L.J.; formal analysis, J.C.W.R. and G.C.L.J.; investigation, J.C.W.R. and G.C.L.J.; resources, J.C.W.R. and G.C.L.J.; data curation, G.C.L.J.; writing—original draft preparation, J.C.W.R.; writing—review and editing, J.C.W.R. and G.C.L.J.; visualization, J.C.W.R. and G.C.L.J.; supervision, J.C.W.R.; project administration, J.C.W.R. All authors have read and agreed to the published version of the manuscript.

**Funding:** This research received no external funding.

**Institutional Review Board Statement:** Not applicable.

**Informed Consent Statement:** Not applicable.

**Data Availability Statement:** Not applicable.

**Conflicts of Interest:** The authors declare no conflict of interest.

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
