# Peer review of "Ordinal Cochran-Mantel-Haenszel Testing and Nonparametric Analysis of Variance: Competing Methodologies"

_stats, doi:10.3390/stats5040056_

Round 1

Reviewer 1 Report

General comment

The paper presents an interesting discussion of the Cochran-Mantel-Haenszel (CMH) test and non-parametric ANOVA while answering the question "which method is preferable?". However, there are several ways I think the paper can be improved before it is accepted for publication.

Here, I divide my comments into what I consider to be "major" issues and "minor" issues. Although, as it is with many "major" issues most of these can be addressed easily enough, whether it be the inclusion of a reference or two or adding a few extra words/sentences to clarify a point.

Major issues

* There is an email address of a Glen Livingston Jr which I assume is a co-author of the paper. However, he is not listed as an author

* While the abstract refers to "ordinal CMH tests" it isn't clear how the methodologies being discussed apply to ordinal variables. So I think some clarity is needed in the abstract concerning the import of ordinal variables to the discussion.

* There are a three instances I found where the text appears as gibberish and is highlighted yellow. See lines 56, 144, 260

* Lines 58-59: The authors state that the tests (OPA to C) become more omnibus with larger degrees of freedom. Could the authors clarify what they mean by "omnibus" and is there a reference that can be included to back up this statement?

* Lines 61-63: It is stated that "the correlation test (C) will have grater power than [the] general association test (OPA) but will have less power than the general association test (GA) for other alternative". Is there a reference or two that can be added here that help the reader gain deeper insight into this comparison?

* To help the reader keep track of which of the four CMH tests are being referred to in Section 2, add the abbreviation when referring to it. For example, when referring to the "correlation test" on line 62 add "(C)" after it.

* Line 78: When talking about the "differences" between the conditional and unconditional CMH tests are they practical, technical, philosophical and/or computational differences. Some insight will into the nature of these differences will help the reader gain a better appreciation for how the conditional and unconditional versions compare

* Line 80: It is on this line that the first mention of scores is made and I think this feature is extremely important. So some mention of how these scores are determined is needed. How does changing these scores impact on the features of the analyses? Is there some literature out there that will help the reader gain an appreciation of the selection of scores.

* Line 84: When discussing the nominal CMH tests are competing unconditional Pearson tests, some clarification is needed on the meaning of "competing". What unconditional Pearson tests test the same hypotheses as the nominal CMH test? If there are competing tests, is there preference to which test is used, or, under what conditions is the nominal CMH test better?

* Line 92: How were the scores 3, 10, 20 and 35, chosen for the income categories? For the job satisfaction categories, why were 1, 3, 4 and 5 chosen? Why not 1,2, 3 and 4, say?

* Line 104: Saying "the p-values are similar but different" doesn't explain anything. Some clarification can be made about how the p-values are "similar" but they they are also "different". If they are different, what is the source of the difference(s)

* Line 109-110: Dealing with categorical explanatory and categorical response variables is interesting. Typically one would not use a Pearson X^2 to test for this sort of association structure since X^2 statistic implies that both variables are explanatory in nature. If an explanatory/response association is the focus then the Goodman-Kruskal tau index is a perfect measure to use. Light and Margolin (1971) found that by rescaling this index then one gets a C-statistic and this is a chi-squared rv with (I-1)(J-1) degrees of freedom. Can the authors discuss whether using the index and its associated C-statistic can be considered in the same context as what is described in the paper. If not, why can't it be considered? Partitions of the tau index and C-statistic also exist for three variables.

* Line 122: what orthogonal polynomials would one use for performing an "unordered NP ANOVA"? Some references would be very helpful here for discussing the role of polynomials. Different orthogonal polynomials are used for different purposes so some clarity about what types of polynomials one would use here is important.

* Line 124: how many orthogonal polynomials are being considered when referring to the "first few"? What is the optimal number of these polynomials. 

* Lines 121 - 130: There are some key points being made here that are important but there isn't a single reference that shows where these points are discussed further. In addition to the above points concerning the orthogonal polynomials, there is no literature that backs up the claim "If the first r - 1 analyses . . . factor".

* Lines 131 - 132: This a two-line paragraph of its own so I wonder . . . what does this statement help suggest about interpretations, applications and extensions to the methodologies that were just discussed (unordered NP ANOVA)

* Line 142: it says "u and v = 1, 2 and 3" but what happens when u/v > 3?

* Line 144: N_{ij} is not defined, although line 69 does refer to \sum_h N_{ihj}. Is this N_{ij}?

* Line 151: I understand what is meant by the (1,2)th and (2,1)th generalised correlations but are they really well known? Perhaps the author(s) can consider adding some references here to help show how well known they are

* Line 165: In what sense is "calculating resampling p-values . . . inconvenient"?

* Line 173: For practical purposes, would it help the reader to refer to the job satisfaction and income variables as the two factors of the "two factor ANOVA"?

* Line 178: Can the authors provide some references that back up the statement "well-known robustness of the ANOVA"?

* Lines 179-181: There is some incomplete information in the text of these lines. Do the authors mean "the gender mean income are $3.91K for females and $4.08K for males"? If so, this doesn't make sense since both appear much higher. If this is not so, then some clarification about the units is needed.

* Line 188: It says "for degree (1, 1) up to degree (3,3)" then nine p-values are given but it isn't clear which p-value refers to which degree. Is the order of the degrees that match with the order of the p-values (1,1), (1,2), (2,1), (2,2), (1,3), (2,3), (3,1), (3,2) and (3,3)?

Minor issues

* Line 33: what is the name of the R package the authors have developed? Is it available on the CRAN? Is there a URL that the reader can be directed to so that they can download the package?

* Line 36: what is meant by "of same"?

* Lines 47-50: include these four tests as dot points

* Line 58: Add "respectively" after "and 1"

* Line 64: should "categories" follow "When the response"?

* Line 66: "The genesis of the CMH methodology appears in the papers of Cochran (1954) . . . and of Mantel and Haenszel (1959). . ."

* Line 88: the chisq.test function is appropriate for two variables. The text implies, but does not directly say, that the analyses considered are for two variables only. So I think some clarity can be made on this.

* Table 1: the formatting of the Job Satisfaction categories is inconsistent. Some are in bold and some are not. Also, the table stretches across two pages which makes it difficult to read.

* Line 90: The data set of Table 1 is from Agresti (2002, Table 2.8 / p. 57) and is a real data set where the source is from the 1996 General Social Survey conducted by the US National Opinion Research Center. Discuss that this is not an artificial data set.

* Line 93: is the CMH general association that Agresti reports the p-values of a OPA test defined on line 47?

* Line 97: concerning "evidence of a correlation", is this correlation of the type C defined on line 50? Is this correlation statistically significant?

* Line 109: Is the "table of counts" a two-way contingency table?

* Lines 117 - 120: There appear to be two bullet points here so make them bullet points

* Line 145: the text here requires indenting

* Line 153: The paragraph starts with "Second" in italics but what is the "First"?

* Line 154: here it refers to "independent" variables but on line 110 reference is made to "explanatory" variables. So some consistency is required about the terminology here

* Line 156: How many sets of orthogonal polynomials are you referring to when you say "multiple sets"?

* Lines 156-157: "A/The new response variable"?

* Line 162: How many are do you mean when it says "Even with [a] few treatments and "several analyses"

* Lines 194-195: It says "insignificant" but do the authors mean "non-statistically significant"?

* Line 203: Do the authors mean "(1,1)th generalised correlation is zero"?

* Line 219: The authors refer to Best and Rayner (2017) for a discussion of a particular example but nothing more is mentioned on this example. Can the authors provide a brief overview of it to help the reader gain a deeper understanding of the point being made here?

* Line 266: All references have "surname, initials" except for Rayner and Beh (2009) which states the second author "surname, first name, initial"

Other points

* There are a range of minor grammatical issues I found (eg missing words) throughout the paper so I ask the author(s) to consider reading through the paper carefully to identify and fix them.

* There is an inconsistent use of UK/Australian and American English used throughout the paper. For example line 31 says has "analyze" but line 86 has "analyse". Also, line 133 says "generalised" but line 195 has "generalized".

* There are other minor comments I have made when reading through the paper but they are very minor and I ask the authors to read  through their paper carefully and correct them before considering a resubmission.

Author Response

Response to the Reviewer 1 Report

All reviewers point out errors that have occurred in the journal’s translation to their format. For example c2 is rendered as gibberish, and random text has its font size changed. Table 1 has multiple introduced deficiencies. There is little the authors can do about these, except resubmit the Word file for journal formatting when we would expect different formatting errors to occur. So we will independently submitted a Word file so such issues can be checked by the journal.

In the Open Reviews the consensus is that the cited references are relevant to the research while all other points can be improved. We have endeavoured to make those improvements, aided particularly by the first reviewer’s extremely helpful feedback, for which we are extremely grateful.

First Reviewer

Major issues

* There is an email address of a Glen Livingston Jr which I assume is a co-author of the paper. However, he is not listed as an author.

Now remedied.

* While the abstract refers to "ordinal CMH tests" it isn't clear how the methodologies being discussed apply to ordinal variables. So I think some clarity is needed in the abstract concerning the import of ordinal variables to the discussion.

Okay. There is an expansion in the text at the former line 80, but we have added a little in the Abstract.

* There are a three instances I found where the text appears as gibberish and is highlighted yellow. See lines 56, 144, 260

These are journal formatting issues.

* Lines 58-59: The authors state that the tests (OPA to C) become more omnibus with larger degrees of freedom. Could the authors clarify what they mean by "omnibus" and is there a reference that can be included to back up this statement?

We have added “See the related discussion in Rayner et al. (2009, page 3).”

* Lines 61-63: It is stated that "the correlation test (C) will have greater power than [the] general association test (OPA) but will have less power than the general association test (GA) for other alternative". Is there a reference or two that can be added here that help the reader gain deeper insight into this comparison?

At the end of this paragraph we’ve “See, for example, the discussion in Rayner et al. (2009, sections 5.3 and 5.4). In this reference the focus is on the power of the Pearson goodness of fit test based on  and the tests based on its components; several examples are given.”

* To help the reader keep track of which of the four CMH tests are being referred to in Section 2, add the abbreviation when referring to it. For example, when referring to the "correlation test" on line 62 add "(C)" after it.

We now refer to the CMH GA and CMH C tests.

* Line 78: When talking about the "differences" between the conditional and unconditional CMH tests are they practical, technical, philosophical and/or computational differences. Some insight will into the nature of these differences will help the reader gain a better appreciation for how the conditional and unconditional versions compare.

Text has been added here.

* Line 80: It is on this line that the first mention of scores is made and I think this feature is extremely important. So some mention of how these scores are determined is needed. How does changing these scores impact on the features of the analyses? Is there some literature out there that will help the reader gain an appreciation of the selection of scores.

We have added “Natural scores (1, 2, 3, …) are frequently chosen, but note the data driven choices made in the job satisfaction example following. Category midpoints or medians also commonly used, as these also reflect the data.” In my experience different sensible scorings make little difference, but I’m reluctant to draw the reader away from the main issues by too frequently delving into literature about other issues. However we return to this in the job satisfaction example.

* Line 84: When discussing the nominal CMH tests are competing unconditional Pearson tests, some clarification is needed on the meaning of "competing". What unconditional Pearson tests test the same hypotheses as the nominal CMH test? If there are competing tests, is there preference to which test is used, or, under what conditions is the nominal CMH test better?

The text has hopefully been clarified. We do not assert that the Pearson tests are better than the nominal CMH tests; indeed we do not know if that is the case. However the availability of software has long been a major issue in the use of statistical methodology, whether or not that is desirable. Indeed used of the CMH tests were boosted when PARCAT became available. See

Landis, J.R., Cooper, M.M., Kennedy, T. and Koch, G.G. (1979). A computer program for testing average partial association in three-way contingency tables (PARCAT). Computer Programs in Biomedicine, 9, 223-246.

* Line 92: How were the scores 3, 10, 20 and 35, chosen for the income categories? For the job satisfaction categories, why were 1, 3, 4 and 5 chosen? Why not 1,2, 3 and 4, say?

Clearly Agresti chose the category scores, no doubt because they reflect the data. Sprent was a strong advocate of data driven statistical methods. As promised above we later look at what differences choosing natural scores over data driven scores makes in the job satisfaction example.

* Line 104: Saying "the p-values are similar but different" doesn't explain anything. Some clarification can be made about how the p-values are "similar" but also "different". If they are different, what is the source of the difference(s)?

The language there was imprecise but, we feel, clear from the context. It has been modified.

* Line 109-110: Dealing with categorical explanatory and categorical response variables is interesting. Typically one would not use a Pearson X^2 to test for this sort of association structure since X^2 statistic implies that both variables are explanatory in nature. If an explanatory/response association is the focus then the Goodman-Kruskal tau index is a perfect measure to use. Light and Margolin (1971) found that by rescaling this index then one gets a C-statistic and this is a chi-squared rv with (I-1)(J-1) degrees of freedom. Can the authors discuss whether using the index and its associated C-statistic can be considered in the same context as what is described in the paper. If not, why can't it be considered? Partitions of the tau index and C-statistic also exist for three variables.

The text is discussing how to construct NP ANOVA tests for multi-factor designs to get at the two dot points between lines 117 and 120 – except that they were lost in the journal mis-formatting. Your insight suggests a parallel way forward in related designs, and that is useful and worthy of investigation at another time.

* Line 122: what orthogonal polynomials would one use for performing an "unordered NP ANOVA"? Some references would be very helpful here for discussing the role of polynomials. Different orthogonal polynomials are used for different purposes so some clarity about what types of polynomials one would use here is important.

Text has been added.

* Line 124: how many orthogonal polynomials are being considered when referring to the "first few"? What is the optimal number of these polynomials.

This is EDA, so there is no optimum. In practice we usually work with r = 3 unless there is reason to not do so, and the text now says this.

* Lines 121 - 130: There are some key points being made here that are important but there isn't a single reference that shows where these points are discussed further. In addition to the above points concerning the orthogonal polynomials, there is no literature that backs up the claim "If the first r - 1 analyses . . . factor".

True. However this is discussed in the forthcoming book, and we’re reluctant to give section numbers in an unpublished work when even at this late stage things may change. Instead we have added “See the examples in Rayner and Best (2018).”

* Lines 131 - 132: This a two-line paragraph of its own so I wonder . . . what does this statement help suggest about interpretations, applications and extensions to the methodologies that were just discussed (unordered NP ANOVA)?

We felt the statement added validation to the methodology.

* Line 142: it says "u and v = 1, 2 and 3" but what happens when u/v > 3?

Twice the text says “usually … ”. As in the examples in Rayner and Best (2018), there is usually little interest in u, v > 3 as interpretation becomes an issue. In fact interpretation is fine for u, v < 2 and for u = 1 (v whatever) and for v = 1 (u whatever). Text has been added here to clarify matters for the reader.

* Line 144: N_{ij} is not defined, although line 69 does refer to \sum_h N_{ihj}. Is this N_{ij}?

Thank you for this. No this is a different context. The text has been adjusted.

* Line 151: I understand what is meant by the (1,2)th and (2,1)th generalised correlations but are they really well known? Perhaps the author(s) can consider adding some references here to help show how well known they are.

You are quite right; some of the intended readers won’t be familiar with umbrella tests. The text has been modified.

* Line 165: In what sense is "calculating resampling p-values . . . inconvenient"?

For many users p-values that can be obtained from standard packages such as JMP or SPSS are preferred, but writing their own code to get resampling p-values is inconvenient or they just don’t have the skills. Even if they do, if the analysis is in some sense preliminary, anything that takes more than the 10 minutes or whatever it takes for resampling code to run, may not be good use of their time. The time taken to write such code will be an added burden.

* Line 173: For practical purposes, would it help the reader to refer to the job satisfaction and income variables as the two factors of the "two factor ANOVA"?

Perhaps not, especially since the paragraph continues to talk about gender and income effects.

* Line 178: Can the authors provide some references that back up the statement "well-known robustness of the ANOVA"?

A reference has been added.

* Lines 179-181: There is some incomplete information in the text of these lines. Do the authors mean "the gender mean income are $3.91K for females and $4.08K for males"? If so, this doesn't make sense since both appear much higher. If this is not so, then some clarification about the units is needed.

Again thank you for this. The values given were incorrect and this have been corrected.

* Line 188: It says "for degree (1, 1) up to degree (3,3)" then nine p-values are given but it isn't clear which p-value refers to which degree. Is the order of the degrees that match with the order of the p-values (1,1), (1,2), (2,1), (2,2), (1,3), (2,3), (3,1), (3,2) and (3,3)?

The detail is now given,

Minor issues

* Line 33: what is the name of the R package the authors have developed? Is it available on the CRAN? Is there a URL that the reader can be directed to so that they can download the package?

Yes, it is available on CRAN and the url is now given.

* Line 36: what is meant by "of same"?

This has been reworded.

* Lines 47-50: include these four tests as dot points.

These are journal formatting issues.

* Line 58: Add "respectively" after "and 1".

Thank you. Done.

* Line 64: should "categories" follow "When the response"?

No. The text is when the responses are ranks … .

* Line 66: "The genesis of the CMH methodology appears in the papers of Cochran (1954)   and of Mantel and Haenszel (1959)."

This has been changed to “are papers by Cochran (1954), … , and by Mantel and Haenszel (1959), …”

* Line 88: the chisq.test function is appropriate for two variables. The text implies, but does not directly say, that the analyses considered are for two variables only. So I think some clarity can be made on this.

This paragraph has been rewritten.

* Table 1: the formatting of the Job Satisfaction categories is inconsistent. Some are in bold and some are not. Also, the table stretches across two pages which makes it difficult to read.

These are journal formatting issues.

* Line 90: The data set of Table 1 is from Agresti (2002, Table 2.8 / p. 57) and is a real data set where the source is from the 1996 General Social Survey conducted by the US National Opinion Research Center. Discuss that this is not an artificial data set.

Thank you. Text has been added here.

* Line 93: is the CMH general association that Agresti reports the p-values of a OPA test defined on line 47?

No, it’s the CMH GA test that he reports. He didn’t give the CMH OPA calculation.

* Line 97: concerning "evidence of a correlation", is this correlation of the type C defined on line 50? Is this correlation statistically significant?

Yes, the text reports a CMH C p-value of 0.0131 and shortly thereafter comments on significance at the 0.05 level.

* Line 109: Is the "table of counts" a two-way contingency table?

Not necessarily, and we don’t want to be explicit, to get bogged down in the detail. We want to get to the two dot points that, as noted previously, were lost in the journal mis-formatting.

* Lines 117 - 120: There appear to be two bullet points here so make them bullet points.

This is a journal formatting issue.

* Line 145: the text here requires indenting.

If you mean requires no indenting, we agree?. Again, this is a journal formatting issue.

* Line 153: The paragraph starts with "Second" in italics but what is the "First"?

At line 141.

* Line 154: here it refers to "independent" variables but on line 110 reference is made to "explanatory" variables. So some consistency is required about the terminology here.

Agreed, so the previous ‘explanatory’ has been changed to ‘independent’.

* Line 156: How many sets of orthogonal polynomials are you referring to when you say "multiple sets"?

As the text says, one set for the response and one for each ordered factor.

* Lines 156-157: "A/The new response variable"?

Thank you. Inserted.

* Line 162: How many are do you mean when it says "Even with [a] few treatments and "several analyses".

Again, being explicit may hinder the reader. This has been changed to “As, for example, in the Job Satisfaction example following, we are proposing … ”. There we are calculating nine generalised correlations, and this illustrates the point.

* Lines 194-195: It says "insignificant" but do the authors mean "non-statistically significant"?

Yes. The text has been changed to “It seems the CMH GA test is not statistically significant because it is testing for too general an alternative: namely generalised correlations up to degree (3, 3). The statistically insignificant higher degree generalised correlations mask the statistically significant (1, 1) generalised correlation”.

* Line 203: Do the authors mean "(1,1)th generalised correlation is zero"?

Yes. Adjusted.

* Line 219: The authors refer to Best and Rayner (2017) for a discussion of a particular example but nothing more is mentioned on this example. Can the authors provide a brief overview of it to help the reader gain a deeper understanding of the point being made here?

The text has been modified to “See the example in Best and Rayner (2017) in which a data set is analysed using Cochran’s Q test. The unconditional test is significant at the 0.05 level, whereas the conditional test is not. The data analyst needs to be aware of the paradigms involved and communicate the conclusions carefully.”. Again we don’t want to get bogged down with the detail.

* Line 266: All references have "surname, initials" except for Rayner and Beh (2009) which states the second author "surname, first name, initial".

Thank you again.

Other points

All noted.

Reviewer 2 Report

The authors compared CMH and NP ANOVA methodologies by stating the properties and characteristics of tests under these two families. The job satisfaction example was used to assess the consistency of p-values across different tests. Recommendations are made from the perspective of test construction.

Major comments: Given the job satisfaction example does not have ground truth, the conclusion that can be drawn from the example is limited. It would be nice to see simulation results which assess the power and type I error (under small sample size) of some of the mentioned tests to compare the strength and weakness of CMH and NP ANOVA. Even one set of scenario for demonstration would be helpful.

Minor comments: Some fonts are larger than the others (line 66 - 71). In table 1, some of the column headers are in bold but others are not. Symbols for chi-square are mis-printed.

Author Response

Second Reviewer

The authors compared CMH and NP ANOVA methodologies by stating the properties and characteristics of tests under these two families. The job satisfaction example was used to assess the consistency of p-values across different tests. Recommendations are made from the perspective of test construction.

Major comments: Given the job satisfaction example does not have ground truth, the conclusion that can be drawn from the example is limited.

It is now noted that

“The data in Table 1, from Agresti (2002), relate job satisfaction and income in males and females. It derives from the 1996 General Social Survey conducted by the US National Opinion Research Center and hence is not artificial data”.

It would be nice to see simulation results which assess the power and type I error (under small sample size) of some of the mentioned tests to compare the strength and weakness of CMH and NP ANOVA. Even one set of scenario for demonstration would be helpful.

Thank you. We feel that such an empirical study should be carefully planned and would be part of subsequent research.

Minor comments: Some fonts are larger than the others (line 66 - 71). In table 1, some of the column headers are in bold but others are not. Symbols for chi-square are mis-printed.

These are journal formatting issues.

Reviewer 3 Report

The author have publication on the subject including book and R package.

It is not clear the value and contribution of the paper in addition to the existing work.

Author Response

Third Reviewer

The author have publication on the subject including book and R package.

It is not clear the value and contribution of the paper in addition to the existing work.

The paper gives a brief overview of the book, and may assist readers with an interest in the methodologies discussed in deciding whether to acquire the book to access aspects of interest.

Reviewer 4 Report

The paper compared several tests for categorical response data. The paper's results are okay. But I feel that the author didn't proofread the manuscript at all. 

1. The review system shows two authors, J.C.W. Rayner * , G.C. LIVINGSTON Jr,  but the manuscript has only one. How many authors does this paper have?  

2. Line 46-49 seem like a single sentence. But why are they separated on different lines?

3. Please clarify what the O^2 distribution is on Line 56.

4. Some words on Line 68-71 have different font sizes. Please make them the same. 

5. Please explain the heightened equal sign on Line 144. 

6. Line 145 should not be indented.  

This list may not be comprehensive. To be considered for publication, careful thorough proofreading is needed. 

Author Response

Fourth Reviewer

The paper compared several tests for categorical response data. The paper's results are okay. But I feel that the author didn't proofread the manuscript at all.

  1. The review system shows two authors, J.C.W. Rayner* , G.C. LIVINGSTON Jr, but the manuscript has only one. How many authors does this paper have?

Two, and the omission of the second author has been remedied.

The issues raised in the subsequent points are all introduced by the journal’s formating process of the submitted Word file.

This list may not be comprehensive. To be considered for publication, careful thorough proofreading is needed.

Round 2

Reviewer 1 Report

My original review consisted of a lot of comments and suggestions for the authors to consider. While not addressing all of them, they have attended to all of the key points I raised and I'm happy to say that, in my view, the paper is acceptable for publication. There remain a few very minor grammatical issues that need fixing but I'm sure they can be picked up during the next stage of production.

Reviewer 2 Report

The authors have addressed my comments.

Reviewer 3 Report

Using "job satisfaction example", this paper compared CMH and NP ANOVA and concluded the results are similar. It is not clear this is enough contribution for publication.

Reviewer 4 Report

The authors have answered all my querstions. I don't have further comments.